# Online communities come with real-world consequences for individuals and societies
Atte Oksanen ⬥ ✉, Magdalena Celuch ⬥ , Reetta Oksa ⬥ & Iina Savolainen ⬥

Online communities have become a central part of the internet. Understanding what motivates users to join these communities, and how they affect them and others, spans various psychological domains, including organizational psychology, political and social psychology, and clinical and health psychology. We focus on online communities that are exemplary for three domains: work, hate, and addictions. We review the risks that emerge from these online communities but also recognize the opportunities that work and behavioral addiction communities present for groups and individuals. With the continued evolution of online spheres, online communities are likely to have an increasingly significant role in all spheres of life, ranging from personal to professional and from individual to societal. Psychological research provides critical insights into understanding the formation of online communities, and the implications for individuals and society. To counteract risks, it needs to identify opportunities for prevention and support.

Online communities are social networks on the internet that utilize technology for interaction. They began to gain popularity in the 1990s with the development of the internet and information and communications technologies[1–3]. The emergence of online communities was further accelerated by Web 2.0 and social media starting in the mid-2000s[4]. Social media platforms provide users fast access to likeminded others, and they speed up communication and offer new ways for interaction[5–7].

The progress of these interactive technologies has been remarkably fast, and they carry both opportunities and risks. The work context is a good example of the complexity of online communities. On one hand, online communication is flexible, fast, and effective regardless of location, and it provides workers with new ways to collaborate and socialize with each other[8]. On the other hand, online communication bears risks, such as workplace cyberbullying[9,10] and misinterpretation of messages and feedback, endangering the mental well-being of employees by inducing technostress, psychological distress, and work exhaustion[11,12]. We recognize that online communities can be supportive and enhance well-being in many ways, but there are also online communities that carry risks for participants and wider society such as hate and addiction communities.

In this perspective article, we review the characteristics of online communities along with the opportunities and risks they present to their users. We cover the role of online communities in the contexts of 1) work, 2) hate and harassment, and 3) addiction, as three domains of outstanding relevance for the society that showcase the multifaceted nature of online

communities. These diverse domains provide an excellent starting point for the theoretical overview of online communities. The first topic of work recognizes that online communication and online communities have a growing importance in today's work life. The second topic concerns research evidence of online hate communities that are based on harmful ideas and actions against other people. This topic has had massive implications on political and societal discussions starting in the 2010s. The third topic talks about online communities in the context of addictions. The online dissemination and proliferation of various views and behaviors have further led to people discovering new, potentially harmful activities or becoming excessively engaged in the digital world. This has prompted significant research into online addictions[13–15].

## How and why online communities form

Human beings are inherently social and seek companionship and social engagement whenever possible[16]. Online communication responds to this social need of belonging. Online communities emerge and thrive in digital spaces, comprised of members who engage in active communication in a shared topic or interest area[17,18]. Online communities can form in a variety of online contexts, including but not limited to social media platforms, discussion forums, and chatrooms[19]. Online communities are significant for finding companionship, fostering connection, accessing information, and receiving support[20,21]. Like any group or community, online communities vary in size, cohesion, and network and focus area. The degree of members'

Faculty of Social Sciences, Tampere University, Tampere, Finland.
✉ e-mail: atte.oksanen@tuni.fi

anonymity may also vary considerably[4,22]. A distinct feature of the global online sphere is that no matter how unusual or rare one's interests are, they are likely to find others with similar interests[23]. Members of online communities are often heterogenous in their social characteristics, including socioeconomic status, life stage, ethnicity, and gender, but tend to be like-minded and homogeneous in terms of their shared interests and attitudes[24,25].

The online sphere provides an ideal environment for the building of networks that hold significance for the social and personal identity of their participants[5]. Social identity theory (SIT), as initially proposed by Tajfel and Turner[26], describes a process in which an individual's identity is partially shaped by their sense of belonging to preferred social groups. This concept is often measured by evaluating an individual's subjective feeling of being a part of the desired groups[27]. Theories and models that use the social identity approach[28,29] are very relevant for understanding online communities and online group behavior.

One of the most important models proposed over the years has been the social identity model of deindividuation effects (SIDE)[30,31]. The model was originally motivated by the topic of online communication's anonymity – an issue that had already drawn the attention of social psychologists in the early 1980s[32]. According to the SIDE model, the deindividuation effect of social identification is especially prevalent in online interactions that are characterized by at least a certain level of anonymity, as it promotes a shift from individual to group self and therefore facilitates behaviors benefiting the group as well as stereotyping outgroup members and viewing them as a representative of their group rather than an individual[33–37]. Relatedly, lack of social cues, such as eye contact, in online interactions has been found to lead to behavioral disinhibition through the so-called online sense of unidentifiability[38].

Context collapse occurs very commonly in social media. This means that boundaries between different social spheres blur together (e.g. interacting with people from different life spheres such as work, family, and friends on the same platform). This can influence and challenge the users' self-presentation and their navigation within different online discussions and audiences[39]. In these diverse social contexts, which can vary greatly in terms of values and norms, users may need to balance their personal authenticity based on their audience expectations. Additionally, they can face situations that compromise their privacy or necessitate self-censorship[40,41]. However, social media simultaneously provides multiple features or opportunities (i.e., affordances) for users to control and maintain their public identities and social networks. In other words, users can manage how they present themselves to the public online and how they interact with their social networks through the tools and functions provided by social media platforms. Typically, these tools include the customization of ones' profile, maintaining visibility, allowing access, editing content, and providing links and connections to other platforms[39,42–44]. Affordances essentially link to the question of how users maintain and create smaller groups or wider communities within certain platforms. This depends on the features and design of each social media platform or internet site.

Characteristics of both internet platforms and applications strongly influence online human behavior. In particular, commercial platforms are designed to attract people's interests in the content. Moral and emotional contents spread faster on social media as they capture users' attention more effectively as compared to neutral content[45]. The process is also facilitated by the algorithms of social media platforms[5,46], and social influence, which lead people to engage with content that is already popular[47]. This has a profound impact on online communities and the way they communicate, especially as these effects have been found to be much stronger within networks comprised of individuals who share similar views than between such networks[48]. Thus, a tendency can be induced toward likeminded individuals who support certain opinions and behaviors, strengthening their existing attitudes and providing a stronger identification with the ingroup[49,50]. In this context, even exposure to differing opinions may primarily serve as a tool to further distinguish between "us" and "them" and hinder productive dialogue[4,51].

These mechanisms are further captured by the Identity Bubble Reinforcement Model (IBRM), which focuses on explaining how characteristics of online communication facilitate the formation of tight-knit networks, namely social media identity bubbles[4,5]. Such bubbles or echo chambers are characterized by three mutually reinforcing features: high identification with the other members (ingroup), homophily or the strong tendency to interact with likeminded others, and information bias, namely heavy reliance on information obtained from the community[4,5]. Involvement in such communities has been found to be associated with compulsive internet use[5], cyberaggression[52] and problem gambling[53]. At the same time, involvement in online identity bubbles facilitates social support and may buffer mental well-being in some situations[54].

In summary, there are three core aspects to consider in online communities. First, technological design is a critical component that impacts what people can do within online communities. The phenomenon of online communities has existed as long as the internet[2], but current social media platforms use different interfaces and AI algorithms than their earlier versions, being essentially engineered to provide content for users. This technological side impacts significantly how people behave and react online. Second, contextual issues are highly important in online communities. Generally, contexts in the online sphere may collapse easily. At the same time, the internet and social media platforms facilitate development of very closed online communities that are based on shared interests. These interests may sometimes be very specific. Core social psychological group theories and their updates provide good tools for understanding evolving online group behavior. SIDE and IBRM are examples of theories that have been proven very useful in empirical research.

## Social media communities at work
The accelerated development of information technology in recent decades has significantly reshaped the workplace. The foundation for this technological advancement was laid in the 1960s with the development of ARPANET, a forerunner to the internet[4]. The same era also witnessed the birth of the Open Diary – an internet-based diary community allowing user participation through messages, thereby serving as a precursor to social media[55]. During the emergence of the internet in the 1990s, known now as the Web 1.0 era, personal web pages, content creation, and numerous work communication tools, such as online telecommunications and email, became prevalent[55,56]. The term Web 2.0 was first coined in 2004, concurrently when Facebook became popular, symbolizing the internet's transition to a more socially diverse and interactive era[55]. The change in user behavior from passive web content consumers to active, bidirectional information creators and editors was an evident part of the transition from Web 1.0 to Web 2.0[4].

In recent years, there has been a substantial increase in the use of digital communication technologies in the workplace, primarily driven by advancements of the internet and social media services[12,57]. Numerous expert organizations are now leveraging corporate social media platforms, such as Microsoft Teams and Workplace from Meta, for their communication needs[8,58]. Networking in enterprise social media platforms facilitates real-time messaging, task organisation, and formal and informal team collaboration, synchronously and asynchronously across organisational groups and in different geographic locations[8,59,60]. Work communication also unfolds through instant-messaging applications, such as WhatsApp, and general social media platforms, such as Facebook, X (formerly Twitter), and LinkedIn, which are being utilised for professional purposes. These social platforms have the potential to encourage professionals to engage in collaboration, share information and ideas, and expand their expertise on a global scale, extending beyond their specific job responsibilities and organisational boundaries[8,61]. Notably, social communication tools have expanded to encompass traditional white-collar environments and now provide value for blue-collar workers as well, for example, as a medium for communication and task organisation[11].

Social media messaging and networking for professional purposes not only enhance knowledge transfer and flow but also nurture the human need

for social belonging[62,63]. Given the growing prevalence of remote and hybrid forms of work, social media has the potential to maintain and foster social interactions regardless of location[57,64]. Remote and hybrid work arrangements can, however, reduce the chances of establishing and nurturing high-quality work relationships[65,66]. Recent studies have also indicated a link between resistance to remote work and having quality workplace relationships[67]. Indeed, working far from the physical work community can increase the growing phenomenon of loneliness at work[65,68,69]. At the same time, in some circumstances, online networking among colleagues nurtures social connections and alleviates feelings of loneliness[54]. Social connections and feelings of belongingness in the work community and one's professional circles are vital to support employees' mental well-being and combat loneliness at work[54,70]. Feelings of loneliness at work can, for example, lower professionals' work engagement, increase their dissatisfaction at work[71] and burnout[72].

The use of social media platforms for professional objectives can enrich communication and foster meaningful connections[8,73]. Professional online relationships can be formed and maintained individually person to person or as a part of bigger professional online communities. In the professional sphere, online communities are commonly referred to as communities of practice due to their origins within the cultural framework of either virtual or traditional organisations[74]. Communication visibility in these online communities of practice can foster knowledge sharing and social learning, trust, and innovation[75–77]. The sense of belonging and togetherness with colleagues can also be enhanced in these online communities[8,78]. Online communities of practice can be a source of affective social support that promotes experiencing group identification and meaningfulness, which in turn can foster employees' engagement in their work[79]. Employees' social media collaboration is also associated with increased team and employee performance[78], and employees perceived social media–enabled productivity[80]. Both formal and informal online communities are known to accelerate professional development[8,81,82].

However, online communities at work can have downsides. These include tensions within the organisation due to employees sharing nonwork-related information that can tighten the bonds and build trust but, interestingly, can also hinder work-related information sharing[76]. Furthermore, stress arising from technology use (i.e., technostress), psychological distress, and burnout are pervasive challenges of professional online collaboration in technologised work environments[11,12]. Concentration problems can emerge, and the boundaries of work and private life can also be blurred and stimulate conflicts[83,84]. In addition, social relationships at work can be challenged[85] by discrimination, ostracism, and face-to-face bullying. These issues are also present in online communication, where they take on new forms and meanings. Work-related cyberbullying[9,10,86] and hate and harassment, which may also come from fellow work community members, can be detrimental for the targets and lead to lowered well-being[87].

## Hate communities

The ease of online communication facilitates the dissemination and proliferation of negative and dangerous views and behaviors. Subsequently, online hate (i.e., cyberhate) and online hate crime have emerged as a prominent area of research in the context of online communication, with the same ease of access contributing to their prevalence[88,89]. Online hate covers a wide range of intensive and hostile actions that target individuals or groups based on their beliefs and demographic factors, such as ideology, sexual orientation, ethnic background, or appearance[90]. The rise of hostile online communication has been considered a growing societal concern over the past decade[4,87,91–93].

The history of hate in online communication goes back to the first internet networks. Organised hate groups have always been interested in the latest technologies to recruit new members and disseminate information. For instance, White supremacists in the US were pioneers in adopting electronic communication networks during the 1980s. Notably, in 1983, neo-Nazi publisher George P. Dietz established the first dial-up bulletin board system (BBS), marking an early utilisation of online communication

methods[94]. Shortly after the inception of the World Wide Web, hate groups marked their online presence. Stormfront.org, launched in 1995, was one of the first and most important hate sites during the Web 1.0 era[95]. Since then, over the past 30 years, continuous technological advancements have significantly enhanced their communication capabilities[4].

Particularly the rise of social media since the mid-2000s was an important game changer in the dissemination and development of online hate. Foxman and Wolf[96](p. 11) summarized this change concerning the Web 2.0 era of social media: "In the interactive community environment of Web 2.0, social networking connects hundreds of millions of people around the globe; it takes just one 'friend of a friend' to infect a circle of hundreds or thousands of individuals with weird, hateful lies that may go unchallenged, twisting mind in unpredictable ways." The last 10 years of the internet have been, however, striking, as online hate has lurked from the margins and started to become a tool of political populists in the Western world[1,97]. Uncertainty of the times with various crises related to terrorism, economy, and the global COVID-19 pandemic have also accelerated the phenomenon.

Research on online hate associated with the COVID-19 pandemic has suggested that, in crisis situations, hate communities can organise quickly and rapidly develop new narratives[98,99], reactively focusing on recent and highly debated issues[100]. Such hateful messages spread most effectively in smaller, hierarchical, and isolated online communities[99], highlighting the dangers of online echo chambers or identity bubbles[4,5,101]. Even if hateful narratives are not endorsed by most users on the platform, the flow of such information tends to be sustained over time, as members of echo chambers encourage each other and amplify their shared worldview[102]. This is often done by referring to and contesting opposing views in a marginalising and undermining way, making counter-messaging ineffective or even counter-effective[103]. Various options of demonstrating (dis)agreement and promoting content on social media are used for creating echo chambers and disseminating hateful content[104]. However, it is worth noting that even on social media sites derived of content-promoting algorithms and vanity metrics present on many of the major platforms, users can quickly learn to recognise and promote extremist content as important and worthy of attention[105].

The example of COVID-19-related hateful activity showed how hate communities effectively spread malicious content across various social media sites, incapacitating moderation attempts of any single platform[98]. Gaming sites are another type of environment where hate and extremist communities organise, recruit, and communicate. It has been argued that the development and characteristics of the gaming industry and the games themselves make online gaming platforms a suitable place for spreading hateful ideologies[106]. Hate communities also commonly use less moderated online spaces as an alternative to mainstream social media platforms, moving toward the creation of parallel ecosystems[107–109]. The need to leave mainstream spaces due to the risk of moderation and censorship is often used for community building by means of leveraging the sense of online persecution and victimisation[109,110].

Hate communities, especially their influential members, use various other techniques and activities for community building. These activities include, for example, the development and promotion of jargon and coded language that underline the "us vs. them" dichotomy, often using derogatory and offensive phrasing[100,104–106,110–113] as well as the use of various audiovisual and interactive materials to capture the recipients' attention[106,107,109]. These strategies can be used differently in different contexts and adapted to groups' needs[114]. The incel (i.e., "involuntarily celibate males") online community is an interesting example of how these strategies are used in practice. According to research, all active participants of online incel discussions commonly use derogatory terms to refer to women[115] and they create powerful dichotomies between themselves and outgroups: both women and society at large, using memes, reels, and other forms of online content to carry their message[115,116].

Another commonly utilised method is ironic and humorous messaging in the form of memes and jokes that further allows for the spread of radical ideologies using seemingly unserious content[104,106,117–119]. Such jokes

and memes are often part of conspiracy talk, which is a type of everyday discourse common among hate communities, referring to conspiracy theories through implicit references and anecdotal evidence from community members' own experiences, often in reaction to news coverage from mainstream sources[119,120]. Research has suggested a strong community-building potential of this type of online discourse, as it allows users to share their concerns and worries and make sense of their experiences[119]. These uncertainties are used by extremist groups to create new anxieties and introduce new problems, as well as to strengthen the community, as evoking feelings of threat can boost the sense of belonging and reinforce the ingroup's worldview[100]. This is especially concerning considering evidence on the associations of supporting far-right ideologies with distrust toward traditional media outlets[121]. Individuals distrustful toward established broadcasters may be motivated to search for alternative sources of information and, as a result, get involved in online hate communities, where they may become further radicalised through community-building practices such as those described above[107].

Research has suggested that, over time, as online communities develop, both positive and negative sentiments in their content increase, and this effect may be stronger in hate communities than in comparable non-hateful groups[111]. This is attributed to the group-formation processes as shared outgroups are established, leading to more negative emotions being expressed. Simultaneously, involvement in a likeminded community results in more positive affect[111]. Interestingly, influential users in online hate communities commonly use seemingly neutral and value-free language, often referring to news from mainstream sources. This is, however, done in a way that is meant to evoke emotion and provoke hateful discussion[122]. This helps to avoid content deletion or user suspension and may further endanger new users looking for alternative sources of information by exposing them to hateful discussions and possibly fostering their radicalisation and involvement in the community[123].

Hateful online content is likely to increase as a result of offline hateful acts[124,125] and local socio-political events that are significant to the group and their worldviews. Together, these can have long-term effects on online hate communities, resulting in increased activity and group cohesion[126]. Although online communities might avoid encouraging offline violence for fear of the discussion being moderated or even completely banned by site administrators[104], they nevertheless contribute to the creation of an environment where hate – both online and offline – is seen as more acceptable and justified[119,127,128]. Indeed, perpetrators of violent extremist acts offline have been previously found to be involved in extremist online communities prior to the act[129], and the spread of hateful content in social media has been tied to subsequent offline hate crimes[93,130].

## Addiction and online communities

There is a complex relationship between addiction and online communities which can be explained through three core factors. First, fast internet connection and mobile devices have enabled unlimited, easy, and continuous online access. Studies have reported that heavy online use symptoms are comparable to substance-related addiction, including mood modification, withdrawal symptoms, conflict, and relapses[15,131]. Second, major social media sites use algorithms to attract and engage their users[4]. Connectedness to others and positive emotions arising from actions and vanity metrics, such as "likes" and supportive comments, reinforce usage and can lead people to become addicted[131,132]. Third, participation in online communities has addictive power. For instance, Naranjo-Zolotov and colleagues[133], who investigated Latin American individuals, found that the sense of a virtual community was the primary factor fueling addiction to social media usage. There is a symbiotic relationship between online communities and technology: technology provides the means for a wide range of activities and it's those activities, rather than the devices themselves, that users typically become addicted to. These online activities often concern the most recognised behavioral addictions such as sex, shopping, gaming or gambling.

When discussing addiction related to online use, it should be acknowledged that, in current terminology, there is a wide variety of terms

expressing the excessive use of the internet or social media. For example, compulsive internet use and problematic internet use are commonly used[134–136]. Technological devices and social media sites are designed to be as engaging as possible. Features such as notifications, personalised content, and interactive elements are strategically implemented to capture users' attention and encourage prolonged usage[137]. These devices and the features within have greatly transformed social interactions, especially in technologically advanced countries and particularly among younger generations who have grown up with smart technology. Current reviews underline a need to build a more complex understanding of different ways of social media use[138]. This involves investigating the geographical, sociocultural, and digital environments within which problematic behaviors arise and unfold[15].

In this Perspective, our focus is on exploring the role of online communities in reinforcing certain problem behaviors. Our examples come from gambling and digital gaming online communities. Online communities centered around gambling and digital gaming are growing in popularity, drawing users to engage and exchange ideas and experiences with others who share similar interests in these activities[21,139]. Online gambling communities usually manifest independently from the actual games, often taking the form of discussion forums dedicated to all aspects of gambling. These forums serve as platforms for participants to engage in dialogues typically including the exchange of tips, strategies, and personal experiences related to gambling[139]. A review of research on online gambling communities indicates that content on these types of online platforms commonly presents gambling in a predominantly positive light[21]. This positive portrayal also seems to resonate with individuals who have a preexisting affinity for gambling, drawing them to participate in the communities online. Joining gambling communities online also appears to be a socially transmitted behavior, as existing members frequently invite their friends or online contacts to join these communities, often through social media where gambling operators also admin and promote communities for their followers[21,140]. The existence of online communities dedicated to gambling provides a convenient platform for gamblers to express interests and emotions they might otherwise hesitate to share in face-to-face interactions. The risk associated with online communities, like those that unite individuals based on a common interest, goals, and norms, is that they might normalise gambling activities and encourage the development of new gambling habits and behaviors. Notably, research has linked active participation in online gambling communities to an increased risk of problem gambling[21,139,141,142].

Online gaming communities are distinct from gambling communities as they inherently exist within the games they are tied to[139]. Virtual social groups that form within games tend to be persistent, and players utilise them to collaborate with each other and enhance their in-game success[143]. Within these communities, members freely exchange skills, knowledge, and virtual assets, including currency used in the game. Players can have different roles and responsabilities within gaming communities. These include sharing responsibilities and communal resources such as in-game items and money[139]. Gaming communities can significantly contribute to the construction of gamers' online identities, which could explain the remarkable success of these communities. This process acts as a validating influence, enabling players to reintegrate themselves through features like avatars and virtual belongings within their communities[144]. Social engagement with fellow players serves as a primary motivator for gaming and can lead to positive social capital gains[145,146], but it can also immerse players in the games, which can lead to excessive time spent on gaming and even to online gaming addiction[147,148]. Further, some in-game activities, such as forms of microtransactions that bear resemblance to gambling, seem to gain support within the gaming community, posing challenges to prevention[149].

Although involvement in various online communities can potentially lead to harmful behaviors and even the initiation or maintenance of addiction, it is crucial to recognise that these communities also serve as a valuable resource for their users. For instance, gamers who harness social bonds within video games often report favorable social outcomes, including

support from in-game friends[150]. Online discussion forums have proven to be a valuable source of support for gamblers, especially those experiencing gambling-related problems or harms[21]. Engaging in conversations online with peers who share similar experiences provides a natural and easily accessible safe space where they can narrate experiences without the fear of judgment. Participants can openly discuss how behaviors like gambling have impacted their lives and share their current self-perceptions. Members of communities focusing on recovery actively exchange information about available resources and offer insights into how to effectively utilise online forums that aid and encourage the recovery process[21,139].

## Growing relevance of online communities

Online communities have growing importance in people's lives today. We are in the middle of remarkable technological change with increasingly ubiquitous computing, which includes major leaps in the development of artificial intelligence technologies and extended realities[151,152]. In some visions, the metaverse is the future of the internet and the 3D model of the internet. The term has been hyped during the early 2020 s, partially so because one of the biggest technology companies, Facebook, renamed itself to Meta and envisioned a metaverse-integrated, immersive ecosystem[152]. Part of the development of the metaverse is tied to technologies and gadgets, but it is hardware independent and functions globally, also within the mobile devices we already use in 2024[152,153]. At this point, it is too early to say how important the metaverse will be in the forthcoming years[154], but it is certain that online communities will play a role in any future development of the internet.

Online communities are fundamentally enabled by the human need for social relatedness[16,155]. Social psychological evidence has shown that group formation takes place easily in any context – also online[2,26,156,157]. This has been shown in both the SIDE and IBRM[4,37]. Characteristics of online communication are tied to the mediated nature of the communication, but, with the help of advanced technologies, the line between on- and offline has become increasingly blurred. Today's research evidence emphasizes the increasing significance of online communities in shaping social connections within both work and everyday life. However, the full extent of this impact is challenging to predict due to the rapid development of internet and social media platforms. Going forward, social psychological theory stands as a cornerstone in understanding the intricate mechanisms of online communities. However, it is crucial to maximise its significance by integrating and considering methodologies and findings from other disciplines of psychology.

In this Perspective, we focused on online communities at work, online hate communities, and online communities based on addiction, and how they contribute to both benefits and risks of human interaction, behavior, and well-being, and what implications such communities hold for the society at large. In the context of work, online communities can facilitate efficient collaboration, knowledge transfer, and social belonging. However, virtual workplace environments may also lead to exclusion, cyberbullying, psychological distress, and technology-induced technostress. Online hate communities pose a worrisome phenomenon, spreading extremist ideas, false information, and conspiracy theories. These activities can have real-world consequences, including increased distrust in institutions and offline deviant behavior. Additionally, online communities related to addiction impact users' time, sleep, relationships, and finances. Despite challenges, online communities offer potential for intervention and support. Research in this multidisciplinary field is urgently relevant, considering technological, societal, and psychological aspects.

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

## Author contributions

Atte Oksanen: conceptualization, project administration, supervision, writing (original draft), writing (review and editing); Magdalena Celuch: conceptualization, writing (original draft), writing (review and editing); Reetta Oksa: conceptualization, writing (original draft), writing (review and editing); Iina Savolainen: conceptualization, writing (original draft), writing (review and editing).

## Competing interests

The authors declare no competing interests.
