## [Peer Review File · Communications Psychology]

29th Feb 24

Dear Atte,

Thank you for your patience during the peer-review process. I am sincerely sorry for the unconscionable delay in returning to you with a decision. Your manuscript titled "Online Communities: Review of social psychological perspectives on theory and research literature" has now been seen by 2 reviewers, and I include their comments at the end of this message.

The reviewers are in principle enthusiastic about your work. However, they also mention a number of concerns. We are very interested in the possibility of publishing your manuscript in Communications Psychology, but would like to consider your revisions in response to these concerns before we make a decision on publication.

In revision, we ask you to pay particular heed to the many constructive remarks that Reviewer #1 makes about the structure of the piece and the need to integrate the disparate sections into an overall framework.

To aid you with that task, I will follow up with editorial comments on a copy of your manuscript within the next two weeks. These comments will provide guidance on how we think the 2 referees' constructive remarks are best utilized to have the piece achieve its full potential.

In sum, we invite you to revise your manuscript taking into account all reviewer and editor comments.

EDITORIAL POLICIES AND FORMATTING

You will find a complete list of formatting requirements following this link:
<https://www.nature.com/documents/commsj-style-formatting-checklist-review-perspective.pdf>

Please use the checklist to prepare your manuscript for resubmission.

* TRANSPARENT PEER REVIEW: Communications Psychology uses a transparent peer review system. This means that we publish the editorial decision letters including Reviewers' comments to the authors and the author rebuttal letters online as a supplementary peer review file. We publish these records for all accepted manuscripts. However, on author request, confidential information and data can be removed from the published reviewer reports and rebuttal letters prior to publication. If your manuscript has been previously reviewed at another journal, those Reviewers' comments would not form part of the published peer review file.

If you have any questions about any of our policies or formatting, please don't hesitate to contact me.

Please use the following link to submit your revised manuscript and a point-by-point response to the referees' comments (which should be in a separate document to any cover letter):

[link redacted]

We hope to receive your revised paper within 16 weeks; if the time required for revision is much longer, we would appreciate it if you could keep us informed about an estimated timescale for resubmission, to facilitate our planning.

Please do not hesitate to contact me if you have any questions or would like to discuss these revisions further. We look forward to seeing the revised manuscript and thank you for the opportunity to review your work.

Best regards,

Marike

Marike Schiffer, PhD

Chief Editor

Communications Psychology

REVIEWERS' EXPERTISE:

Reviewer #1: online communities/online behaviour / group behaviour

Reviewer #2 online communities/online behaviour

REVIEWERS' COMMENTS:

Reviewer #1 (Remarks to the Author):

Thank you for the opportunity to review the paper “Online Communities:Review of social psychological perspectives on theory and research literature”. The paper presents a narrative review of three types of online communities: work online communities, online hate communities, and online communities based on addiction. The review covers a lot of significant research on these specific online communities, but not being a systematic review, some recent and highly relevant work is, not surprisingly, missing (see my recommended references below). The review integrates well some classic and more recent theoretical arguments, and there is of course the value of bringing into the focus the topic of high public interest – online communities. However, in my opinion the current version of the paper could be improved by addressing the following issues.

1. The title of the review is too broad, but this perhaps reflects a more general issue with the review – even if it is informative and quite comprehensive, the question/s driving this review is not clear. Is it about the role of these online communities on their members, or the society at large? This a very general question, but either way, the authors should provide a more focused analysis and interpretation of the type of answers we get from the research reviewed. If we look at various theories relevant here, can we come up with a framework, a model, or a set of propositions, that would help us explain in a more nuanced way how these communities affect their members, and potential risks and/or benefits of engaging with such communities.

2. It is not clear why the review focuses on these three types of communities and not others? Is there something to gain from bringing together these three types of online communities and include them in one review? Do they share underpinning processes or are they distinct in ways that would help the field better understand the socio-psychological dynamics and evolution of online communities. The review should include a section that integrates the key points discussed for each

type of community, highlighting how the field can benefit from the perspective provided by the review (almost like a Discussion section in an empirical paper, before the Conclusion).

3. From the three sections on online communities, the section of online communities based on addition, was the most confusing to me. Initially when behavioural addictions were mentioned, I thought that these online communities would also include communities based on sharing various behavioural addictions (e.g., online shopping, porn, etc.), as opposed to substance addictions. I could not follow how the discussion on smart phone and technology addiction is part of the argument. I would recommend a more systematic approach to the sections on online communities. For example, I would start with a clear definition or description of what is meant by these communities (also mentioning if the broader category of online community covers a range of different sub-categories such as in the case of addiction community including gambling and gaming sub-categories). Next the relevant research and theory can be described, but this descriptive part should always be followed by an integrative section where the key points of the sub-section are reiterated briefly (addressing the “so what?” question).

4. The concluding paragraph (i.e., starting with “Online communities – like any other communities – can have both positive and negative sides. Some communities are supportive and beneficial, and some portray hazards for individual well-being (...)” is far too general. I recommend replacing it with a paragraph more concisely summarising the what the review did, and more directly discuss the implications and the importance of the review’s findings/propositions for the cross-disciplinary study of online communities. Most people in the field would already agree that the study of online communities is increasingly relevant, so more specific points about the contribution to the field would make this section of the paper much stronger.

Overall, this is a timely and highly needed paper in the field, and by sharpening the argument and taking further the implications and significance of the research discussed in the review, the potential impact of the review can be enhanced.

Some references in relation to research on the online hate communities:

Gaudette, T., Scrivens, R., Davies, G., & Frank, R. (2021). Upvoting extremism: Collective identity formation and the extreme right on Reddit. *New Media & Society*, 23(12), 3491-3508.

Scrivens, R., Davies, G., & Frank, R. (2020). Measuring the evolution of radical right-wing posting behaviors online. *Deviant Behavior*, 41(2), 216-232.

Scrivens, R. (2021). Exploring radical right-wing posting behaviors online. *Deviant Behavior*, 42(11), 1470-1484.

Hutchinson, J., Amarasingam, A., Scrivens, R., & Ballsun-Stanton, B. (2023). Mobilizing extremism online: comparing Australian and Canadian right-wing extremist groups on Facebook. *Behavioral Sciences of Terrorism and Political Aggression*, 15(2), 215-245.

Törnberg, P., & Törnberg, A. (2022). Inside a White Power echo chamber: Why fringe digital spaces are polarizing politics. *new media & society*, 14614448221122915.

Bright, J., Marchal, N., Ganesh, B., & Rudinac, S. (2022). How do individuals in a radical echo chamber react to opposing views? Evidence from a content analysis of Stormfront. *Human Communication Research*, 48(1), 116-145.

Reviewer: Dr Ana-Maria Bliuc

Reviewer #2 (Remarks to the Author):

The article deals with an interesting topic, online communities and the state of the art of academic literature on this subject from a social psychological perspective, which fits both Journal aims and scopes and readers' interests.

The article is well structured and well written, and it successfully guides the readers through the authors' arguments.

The goals of the paper, as well as the rationale of the

study, are clearly formulated and are very interesting and useful from the point of view of the relevant disciplinary field.

The literature review is relevant and up-to-date; although it could be expanded to some topics and authors that I will discuss shortly.

It is important to highlight the unbiased authors' stance on the online communities' role in our contemporary society: they provide both positive and negative views about the phenomenon.

Obviously, the innovative potential is not high, but it is generally not the main focus of articles reviewing the literature on a scientific topic. However, the fair and unbiased perspective adopted in dealing with the topic, coupled with the fact that an up-to-date review is always something 'new' in a certain sense, guarantees a certain amount of innovativeness.

However, there are some aspects that, in my opinion, could be addressed or deepened.

1) In the section "Online communities: phenomenon and core theories" I think that the authors should also consider some works from scholars that helped studying and shaping the idea of online communities like Jenkins, Levy (collective intelligence paradigm), Mafessoli. I would also consider some contributions from the Marketing field like the ones proposed by Cova and Kozinets;

2) When discussing about the impact of social media on the online communities (especially about the problems they may cause, like the echo chamber you correctly mentioned), I think that the idea of the "context collapse" introduced by Danah Boyd could also be considered;

3) I would also consider the impact of the affordances of different social media platforms in shaping the way a person may build the personal identity when acting in an online community;

4) As regard the hate communities, I think an interesting example that you can address is the Incel community.

COMMSPSYCHOL-23-0349

We would like to thank the reviewers for their time and effort in commenting on the manuscript. We have now addressed all the comments. The corrections now provide better integration of different perspectives reflecting comments by reviewers.

Submission includes both a clean version of the manuscript and a version showing changes. We have highlighted the revisions in the text in yellow and marked the reference to the specific reviewer comment using color codes: green for Reviewer 1 (R1), blue for Reviewer 2 (R2). For example, R1: 2 refers to comment 2 by Reviewer 1 in the correction log. Please see detailed responses to each comment below.

REVIEWERS' EXPERTISE:

Reviewer #1: online communities/online behaviour / group behaviour
Reviewer #2 online communities/online behaviour

REVIEWERS' COMMENTS:

Reviewer #1 (Remarks to the Author):

Thank you for the opportunity to review the paper “Online Communities:Review of social psychological perspectives on theory and research literature”. The paper presents a narrative review of three types of online communities: work online communities, online hate communities, and online communities based on addiction. The review covers a lot of significant research on these specific online communities, but not being a systematic review, some recent and highly relevant work is, not surprisingly, missing (see my recommended references below). The review integrates well some classic and more recent theoretical arguments, and there is of course the value of bringing into the focus the topic of high public interest – online communities. However, in my opinion the current version of the paper could be improved by addressing the following issues.

1. The title of the review is too broad, but this perhaps reflects a more general issue with the review – even if it is informative and quite comprehensive, the question/s driving this review is not clear. Is it about the role of these online communities on their members, or the society at large? This a very general question, but either way, the authors should provide a more focused analysis and interpretation of the type of answers we get from the research reviewed. If we look at various theories relevant here, can we come up with a framework, a model, or a set of propositions, that would help us explain in a more nuanced way how these communities affect their members, and potential risks and/or benefits of engaging with such communities.

ANSWER [R1:1]: Thank you for your thoughtful comments. We have now revised both the article and the title. We consider that the main title is actually very good and it has high search engine potential. We have revised the subtitle. This is a perspectives article and we do provide exactly that, perspectives for online communities. The text has been

gone through major revisions, we have suggested all the literature and also considered all critical points.

2. It is not clear why the review focuses on these three types of communities and not others? Is there something to gain from bringing together these three types of online communities and include them in one review? Do they share underpinning processes or are they distinct in ways that would help the field better understand the socio-psychological dynamics and evolution of online communities. The review should include a section that integrates the key points discussed for each type of community, highlighting how the field can benefit from the perspective provided by the review (almost like a Discussion section in an empirical paper, before the Conclusion).

ANSWER [R1: 2] Thank you for your comment. We have now provided additional information about the reasons for selecting these three topics or perspectives to online communities. We believe these are very important examples that help us to provide a coherent view of online communities.

3. From the three sections on online communities, the section of online communities based on addition, was the most confusing to me. Initially when behavioural addictions were mentioned, I thought that these online communities would also include communities based on sharing various behavioural addictions (e.g., online shopping, porn, etc.), as opposed to substance addictions. I could not follow how the discussion on smart phone and technology addiction is part of the argument. I would recommend a more systematic approach to the sections on online communities. For example, I would start with a clear definition or description of what is meant by these communities (also mentioning if the broader category of online community covers a range of different sub-categories such as in the case of addiction community including gambling and gaming sub-categories). Next the relevant research and theory can be described, but this descriptive part should always be followed by an integrative section where the key points of the sub-section are reiterated briefly (addressing the “so what?” question).

ANSWER [R1:3]: Thank you for this valuable feedback. We have largely re-written this section to enhance clarity and focus. We now emphasize the connection between technological developments and addictive behaviors. But the main focus of the section is on gambling and gaming communities.

4. The concluding paragraph (i.e., starting with “Online communities – like any other communities – can have both positive and negative sides. Some communities are supportive and beneficial, and some portray hazards for individual well-being (...)” is far too general. I recommend replacing it with a paragraph more concisely summarising the what the review did, and more directly discuss the implications and the importance of the review’s findings/propositions for the cross-disciplinary study of online communities. Most people in the field would already agree that the study of online communities is increasingly relevant, so more specific points about the contribution to the field would make this section of the paper much stronger.

ANSWER [R1:4]: Thank you for this recommendation. We agree that highlighting specific contributions of the review is necessary and strengthens this part. We have revised the paragraph to provide a clearer and more focused conclusion, emphasizing our findings and their significance in the context of the broader field.

Overall, this is a timely and highly needed paper in the field, and by sharpening the argument and taking further the implications and significance of the research discussed in the review, the potential impact of the review can be enhanced.

Some references in relation to research on the online hate communities:

Gaudette, T., Scrivens, R., Davies, G., & Frank, R. (2021). Upvoting extremism: Collective identity formation and the extreme right on Reddit. *New Media & Society*, 23(12), 3491-3508.

Scrivens, R., Davies, G., & Frank, R. (2020). Measuring the evolution of radical right-wing posting behaviors online. *Deviant Behavior*, 41(2), 216-232.

Scrivens, R. (2021). Exploring radical right-wing posting behaviors online. *Deviant Behavior*, 42(11), 1470-1484.

Hutchinson, J., Amarasingam, A., Scrivens, R., & Ballsun-Stanton, B. (2023). Mobilizing extremism online: comparing Australian and Canadian right-wing extremist groups on Facebook. *Behavioral Sciences of Terrorism and Political Aggression*, 15(2), 215-245.

Törnberg, P., & Törnberg, A. (2022). Inside a White Power echo chamber: Why fringe digital spaces are polarizing politics. *new media & society*, 14614448221122915.

Bright, J., Marchal, N., Ganesh, B., & Rudinac, S. (2022). How do individuals in a radical echo chamber react to opposing views? Evidence from a content analysis of Stormfront. *Human Communication Research*, 48(1), 116-145.

ANSWER: Thank you for these suggestions. We have now cited them all in our article and also expanded the literature.

Reviewer: Dr Ana-Maria Bliuc

Reviewer #2 (Remarks to the Author):

The article deals with an interesting topic, online communities and the state of the art of academic literature on this subject from a social psychological perspective, which fits both Journal aims and scopes and readers' interests.

The article is well structured and well written, and it successfully guides the readers through the authors' arguments.

The goals of the paper, as well as the rationale of the study, are clearly formulated and are very interesting and useful from the point of view of the relevant disciplinary field. The literature review is relevant and up-to-date; although it could be expanded to some topics and authors that I will discuss shortly.

It is important to highlight the unbiased authors' stance on the online communities' role in our

contemporary society: they provide both positive and negative views about the phenomenon. Obviously, the innovative potential is not high, but it is generally not the main focus of articles reviewing the literature on a scientific topic. However, the fair and unbiased perspective adopted in dealing with the topic, coupled with the fact that an up-to-date review is always something 'new' in a certain sense, guarantees a certain amount of innovativeness.

However, there are some aspects that, in my opinion, could be addressed or deepened.

1) In the section "Online communities: phenomenon and core theories" I think that the authors should also consider some works from scholars that helped studying and shaping the idea of online communities like Jenkins, Levy (collective intelligence paradigm), Maffessoli. I would also consider some contributions from the Marketing field like the ones proposed by Cova and Kozinets;

ANSWER [R2: 1]: Thank you for your comment. We have revised this section and deepened the discussion. Collective intelligence is a major topic and would deserve an article of its own. This type of article would have a good fit with theories by Maffessoli, Levy and others. However, we believe it is a bit out of the scope of our article.

2) When discussing about the impact of social media on the online communities (especially about the problems they may cause, like the echo chamber you correctly mentioned), I think that the idea of the "context collapse" introduced by Danah Boyd could also be considered;

ANSWER [R2:2]: Thank you for this valuable comment. We have discussed the concept of context collapse now in the article.

3) I would also consider the impact of the affordances of different social media platforms in shaping the way a person may build the personal identity when acting in an online community;

ANSWER [R2:3]: Thank you for noting this. We have now discussed affordances in the the theory section. We believe revisions made improve the article.

4) As regard the hate communities, I think an interesting example that you can address is the Incel community.

ANSWER [R2:4]: Thank you for this suggestion. We have now included this example.

10th May 24

Dear Atte,

Your Perspective Article titled "Online Communities: Social psychological perspectives on theory and empirical research" has now been seen by 1 of the previous referees, whose comments appear below. In the light of their advice I am delighted to say that we are happy, in principle, to publish it in *Communications Psychology* under a Creative Commons 'CC BY' open access license.

We will not send your revised paper for further review if all editorial requests, which are informed by the referees' comments, have been addressed. If the revised paper is in *Communications Psychology* format, in accessible style and of appropriate length, we shall accept it for publication immediately. I have attached an edited version of your manuscript, and ask you to attend to each comment in detail.

EDITORIAL REQUESTS:

* Please review the changes in the attached copy of your manuscript, which has been edited for style, and address the comments and queries I have added. If using Word, please use the 'track changes' feature to make the process of accepting your manuscript more efficient.

* *Communications Psychology* uses a transparent peer review system. On author request, confidential information and data can be removed from the published reviewer reports and rebuttal letters prior to publication. If you are concerned about the release of confidential data, please let us know specifically what information you would like to have removed. Please note that we cannot incorporate redactions for any other reasons.

*If you have not done so already, please alert me to any related manuscripts from your group that are under consideration or in press at other journals, or are being written up for submission to other journals (see www.nature.com/authors/editorial_policies/duplicate.html for details).

FORMATTING GUIDELINES:

You will find a complete list of formatting requirements following this link:

<https://www.nature.com/documents/commsj-style-formatting-checklist-review-perspective.pdf>

Please use the checklist to prepare your manuscript for final submission. In the following, I also highlight some issues of particular importance.

In order to accept your paper, we require the following:

- * A cover letter describing your response to our editorial requests.

- * A separate document detailing your point-by-point response to any issues raised by our referees (please include the referees' comments in this document).

- * The final version of your text as a Word or TeX/LaTeX file, with any tables prepared using the Table menu in Word or the table environment in TeX/LaTeX and using the 'track changes' feature in Word.

At acceptance, the corresponding author will be required to complete an Open Access Licence to Publish on behalf of all authors, declare that all required third party permissions have been obtained.

Please note that your paper cannot be sent for typesetting to our production team until we have received this information; therefore, please ensure that you have this ready when submitting the final version of your manuscript.

ORCID

Communications Psychology is committed to improving transparency in authorship. As part of our efforts in this direction, we are now requesting that all authors identified as 'corresponding author' create and link their Open Researcher and Contributor Identifier (ORCID) with their account on the Manuscript Tracking System (MTS) prior to acceptance. ORCID helps the scientific community achieve unambiguous attribution of all scholarly contributions. For more information please visit <http://www.springernature.com/orcid>

For all corresponding authors listed on the manuscript, please follow the instructions in the link below to link your ORCID to your account on our MTS before submitting the final version of the manuscript. If you do not yet have an ORCID you will be able to create one in minutes.

IMPORTANT: All authors identified as 'corresponding author' on the manuscript must follow these instructions. Non-corresponding authors do not have to link their ORCIDs but are encouraged to do so. Please note that it will not be possible to add/modify ORCIDs at proof. Thus, if they wish to have their ORCID added to the paper they must also follow the above procedure prior to acceptance.

To support ORCID's aims, we only allow a single ORCID identifier to be attached to one account. If you have any issues attaching an ORCID identifier to your MTS account, please contact the Platform Support Helpdesk.

[link redacted]

We hope to hear from you within two weeks; please let us know if the process may take longer.

Best regards,

Marike

Marike Schiffer, PhD

Chief Editor

Communications Psychology

REVIEWERS' COMMENTS:

Reviewer #1 (Remarks to the Author):

Thank you for addressing the points I made in relation to the first version of the paper. I feel that most of my points have been addressed well in the revision. The revised version would still benefit from a careful proof reading as there are still some small punctuation and spelling errors. One last “deep editing” for clarity would be also helpful.

In relation to my initial point about how the three types of online communities fit together, perhaps it is worth making the link between the communities and the key areas of psychology that they would fall under as well as the broader domains: organisational psychology (the work domain), political/social psychology (inclusion, psychological well-being, and safety), and health psychology (addictive behaviours). This point is picked up again in the Conclusion which is very good, but I still feel that this point might better highlight the need to study online communities by using different (theoretical) perspectives and disciplinary lens. For consistence, I would recommend slightly changing the point about the importance of social psychological perspectives in the study of addiction – that is, I think it is important to highlight the importance of social psychology theory to understand the underpinning mechanisms in these communities, at the same time making sure we integrate and consider methodology and respectively research findings from other (sub) disciplines/areas of Psychology. Finally, a small issue: I had expected that some of references I suggested that would be incorporated in the argument rather than just added as in text citations.